# Serum IL-13 Predicts Response to Golimumab in Bio-Naïve Ulcerative Colitis

**DOI:** 10.3390/jcm11174952

**Published:** 2022-08-23

**Authors:** Naohiko Kinoshita, Kazuki Kakimoto, Hikaru Shimizu, Koji Nishida, Keijiro Numa, Yuka Kawasaki, Hideki Tawa, Kei Nakazawa, Ryoji Koshiba, Yuki Hirata, Naokuni Sakiyama, Eiko Koubayashi, Toshihisa Takeuchi, Takako Miyazaki, Kazuhide Higuchi, Shiro Nakamura, Hiroki Nishikawa

**Affiliations:** 2nd Department of Internal Medicine, Osaka Medical and Pharmaceutical University, 2-7 Daigakumachi, Takatsuki City 569-8686, Osaka, Japan

**Keywords:** ulcerative colitis, IL-13, golimumab

## Abstract

A certain number of patients with ulcerative colitis (UC) are refractory to anti-TNF-α antibodies; biomarkers are thus needed to predict treatment efficacy. This study aimed to evaluate whether serum biomarkers that were reported to be associated with UC or anti-TNF-α antibody could predict the response to golimumab, a human anti-TNF-α monoclonal antibody, in bio-naïve patients with UC. We prospectively enrolled 23 consecutive patients with UC who were treated with golimumab. Serum samples were collected before the first golimumab dose. Eleven molecules were measured by electrochemiluminescence (ECL) or enzyme-linked immunosorbent assay (ELISA) and their association with efficacy after 10 weeks of golimumab treatment. Among the serum biomarkers, IL-13 levels were significantly higher in the non-remission group than in the remission group (*p* = 0.014). IL-15 levels were significantly lower in the non-response group than in the response group (*p* = 0.04). For clinical remission at week 10, the IL-13 0.20 concentration of pg/mL was associated with a sensitivity and specificity of 82.4% and 83.3%, respectively. Serum IL-13 may be a biomarker to predict golimumab efficacy in biologic-naïve patients with UC, and thus may help to tailor personalized treatment strategies.

## 1. Introduction

In inflammatory bowel diseases (IBDs), such as UC and Crohn’s disease (CD), immune cells, including T cells and macrophages, are activated in intestinal tissues, resulting in the production of inflammatory cytokines and chronic inflammation [1]. CD is often thought to be a prototype of a type 1 helper T cell (Th1)-mediated disease because the primary inflammatory mediators are Th1 cytokines such as IFN-γ, TNF-α, and IL-12, whereas UC is usually thought to be a type 2 helper T cell (Th2)-mediated disease because of the increased intestinal expression of Th2 cytokines such as IL-4, IL-5, and IL-13 [2,3]. Recently, various molecular-targeted drugs have been developed to treat IBD, and the choice of therapeutic agents in actual clinical practice has become extremely complex [4]. In general, therapeutic agents are selected based on factors such as disease severity and drug safety, but it would be of great clinical significance if biomarkers which can predict the efficacy of therapeutic agents in advance according to individual pathological conditions, such as the cytokine profile of the patient, were available. The TNF family, represented by TNF-α, plays an important role in innate and adaptive immunity and is also an important UC cause [5]. Anti-TNF-α antibodies such as infliximab, adalimumab, and golimumab (GLM) have been reported to be highly effective against UC [6,7,8,9]. However, there are not a few cases in which anti-TNF-α antibodies are ineffective. Therefore, it is important to predict the response to the anti-TNF-α antibody. Several studies have reported biomarkers for predicting the efficacy of the anti-TNF-α antibody. Oncostatin M (OSM) is a member of the IL-6 cytokine family with gp130 as a common receptor subunit [10]; OSM gene expression in intestinal mucosal tissues has been reported to be a predictor of the efficacy of anti-TNF-α antibody preparations in patients with IBD [11]. In CD, IL-13Rα2 gene expression in the intestinal mucosal tissues has also been reported to be associated with the efficacy of the anti-TNF-α antibody [12]. However, analysis of intestinal mucosal tissue requires biopsy by colonoscopy, which is invasive. Therefore, we believe that serum-based biomarkers, which are simpler and less invasive, are useful in clinical practice. While several studies have examined the relationship between gene expression in intestinal mucosal tissues and the efficacy of anti-TNF-α antibody preparations, few studies have examined protein expression in serum. Additionally, since the expression pattern of inflammatory cytokines changes with the type of drug after administration of some biological agents, it is preferable to use serum from bio-naïve UC patients to more clearly analyze the association between the expression of molecules in UC and the efficacy of anti-TNF-α antibodies. In this study, we measured molecules reported to be associated with UC or anti-TNF-α antibody preparations in the serum of UC patients who had never used biologic therapies to identify potential biomarkers to predict GLM efficacy.

## 2. Materials and Methods

### 2.1. Patients

This study was conducted at the Second Department of Internal Medicine, Osaka Medical and Pharmaceutical University, from May 2017 to October 2020. This study included consecutive patients with UC who were initiated on GLM therapy and fulfilled the following inclusion criteria: age between 18 and 75 years with a diagnosis of moderate-to-severe and biologically naïve UC. Exclusion criteria included the impending need for surgery; history of malignancy; and contraindications as specified in the GLM product monogram, such as tuberculosis, severe infection, and congestive heart failure. GLM was administered subcutaneously at an initial dose of 200 mg and 100 mg at week 2. Thereafter, 100 mg of GLM was administered subcutaneously every four weeks. This study was performed following the principles of the Declaration of Helsinki. The study protocol was approved by the ethics committee of Osaka Medical and Pharmaceutical University. Written informed consent was obtained from all patients included in this study.

### 2.2. Outcomes

Clinical remission was defined as a *p* Mayo score of ≤2 and a Mayo score of ≤1. Clinical response was defined as a 30% or greater reduction in the Mayo score from the enrolment time. The efficacy was evaluated after 10 weeks of GLM administration.

### 2.3. Measurement of Serum Biomarkers

Serum samples were collected immediately before and 6 weeks after the first GLM dose. Eleven molecules were measured using ECL or ELISA.

### 2.4. ECL for IL-1β, IL-5, IL-13, IL-15, IL-33, IL-12/IL-23p40, Eotaxin-1

The assay was performed using the V-PLEX Custom Human Biomarkers Kit (K15050D) from MSD (Rockville, MD, USA). The detection sensitivity was 0.05 pg/mL for IL-1β, 0.14 pg/mL for IL-5, 0.24 pg/mL for IL-13, 0.15 pg/mL for IL-15, 0.59 pg/mL for IL-33, 0.33 pg/mL for IL-12/IL-23 p40, and 3.2 pg/mL for eotaxin-1.

### 2.5. ELISA for OSM, IL-4, IL-13Rα2, IL-17

Serum OSM levels were measured by ELISA using a commercial kit (ELH-OSM) from Ray Biotech (Peachtree Corners, GA, USA). Serum IL-4 level was measured by ELISA using a commercial kit (ab215089) from Abcam (Cambridge, UK). Serum IL13Rα2 was measured by ELISA using a commercial kit (SEB643Hu) from Cloud-Clone (Katy, TX, USA). Serum IL-17 level was measured by ELISA using a commercial kit (ab119535) from Abcam (Cambridge, UK). The detection sensitivity was 1.0 pg/mL, 1.08 pg/mL, 13.8 pg/mL, and 0.5 pg/mL, respectively.

### 2.6. Statistical Analysis

All statistical analyses were performed using JMP v15.2.1 software (SAS Institute, Cary, NC, USA). Quantitative data were summarized using the median and interquartile range (IQR), and categorical variables were described using frequency and percentage. We used the Wilcoxon signed-rank test to compare serum IL-1β, IL-5, IL-13, IL-15, IL-33, IL-12/IL-23p40, eotaxin-1, OSM, IL-4, IL-13Rα2, and IL-17 levels between patients who achieved and did not achieve the specified efficacy outcomes. Receiver operating characteristic (ROC) curves were constructed to determine the best molecule sensitivity and specificity cut-off values at early time points for predicting outcomes at week 10. Kaplan–Meier survival analysis plots and log-rank tests were used to compare GLM termination rates between the study arms. Statistical significance was set as *p* < 0.05 (two-sided test).

## 3. Results

### 3.1. Patient Characteristics

Twenty-three patients were enrolled after the inclusion and exclusion criteria were applied. The study included 13 men and 10 women (Table 1). The median age was 47 years, and the UC median duration was 1.2 (0.4–18) years. The proportion of UC patients with an extensive disease was 17 (74%); eight (35%) were taking corticosteroids at the start of GLM treatment, and six (26%) were taking immunosuppressive drugs as concomitant therapy. The median C-reactive protein (CRP) was 0.52 (0.01–8.17) mg/dL and the median *p* Mayo score was 7 (3–9).

### 3.2. Efficacy Outcomes and Clinical Characteristics Associated with Clinical Remission

Ten weeks after GLM initiation, 6 patients (26%) were in clinical remission and 10 patients (43%) were in clinical response. The corresponding dose of corticosteroids in the remission group was 3.5 mg (0–20) at baseline, ~0 mg (0–5) at week 10, and 0 mg at week 52. In the non-remission group, the corresponding corticosteroid dose was 0 mg (0–10) at baseline, 0 mg (0–1) at week 10, and 0 mg (0–5) at week 52. We examined the clinical factors associated with clinical remission at week 10. As shown in Table 2, age was significantly higher in the non-remission group. No other baseline factors were associated with clinical remission at week 10.

### 3.3. Relationship between Serum Molecules and Efficacy Outcomes

In serum samples before the first GLM dose, serum levels of molecules reported to be associated with anti-TNFα antibody and UC were measured and examined for association with efficacy after 10 weeks of GLM treatment.

IL-13 levels were significantly higher in the non-remission group than in the remission group (median (IQR) (0–2.18) 0.475 pg/mL vs. (0–0.378) 0 pg/mL, *p* = 0.014, Figure 1E). IL-13 level was also higher in the non-response group than in the response group but the difference was not significant (median (IQR) (0–2.18) 0.475 pg/mL vs. (0–0.817) 0.20 pg/mL, *p* = 0.069, Figure 2E). IL-13Rα2 was not associated with GLM efficacy (median (IQR) (0.079–0.231) 0.098 pg/mL vs. (0.075–0.803) 0.11 pg/mL, *p* = 0.327, Figure 1F). IL-15 levels did not differ between the remission and non-remission groups but were significantly higher in the response group than in the non-response group (median (IQR) (2.37–6.68) 3.91 pg/mL vs. (2.15–4.85) 2.96 pg/mL, *p* = 0.04, Figure 2G). OSM was not associated with GLM efficacy (median (IQR) (14.1–1119.5) 28.3 pg/mL vs. (13.2–1090.6) 16.5 pg/mL, *p* = 0.484, Figure 1J). No other significant differences were observed for IL-1β, IL-4, IL-5, IL-12/23 p40, IL-17, IL-33, and eotaxin-1 between the remission and non-remission groups or between the response and non-response groups (Figure 1 and Figure 2).

### 3.4. ROC Curve Analyses

ROC curves were developed to identify the optimal threshold for serum IL-13 and IL-15 associated with GLM efficacy. The area under the ROC curve (AUC) for IL-13 of clinical remission at week 10 was 0.84 (95% confidence interval (CI), 0.68–1) and had moderate accuracy (0.7–0.9) (Figure 3A). For clinical remission at week 10, the IL-13 concentration of 0.20 pg/mL was associated with a sensitivity and specificity of 82.4% and 83.3%, respectively. The AUC for IL-15 of clinical response at week 10 was 0.75 (95% confidence interval (CI), 0.54–0.97) and had moderate accuracy (Figure 3B). For clinical response at week 10, the IL-15 concentration of 3.03 pg/mL was associated with a sensitivity and specificity of 88.9% and 64.3%, respectively.

### 3.5. Kaplan–Meier Curve Analyses

Figure 4 shows the survival analysis of the time to discontinuation of GLM treatment for up to 54 weeks after GLM administration. The GLM termination rate by week 54 was significantly lower in the serum IL-13 < 0.20 pg/mL group than in the >0.20 pg/mL group (*n* = 8 34% vs. *n* = 15 66%, respectively; *p* = 0.0247).

### 3.6. Efficacy of GLM and Changes over Time

Figure 5 shows serum IL-13 levels before and 6 weeks after GLM administration in the remission/non-remission (Figure 5A) and response/non-response (Figure 5B) groups. There was no significant change in IL-13 before and 6 weeks after GLM administration in either group.

### 3.7. Correlation of IL-13 with Other Factors

To determine the clinical background of UC with high IL-13 levels, we analyzed the correlation between IL-13 levels and clinical factors. No correlation was found between IL-13 and CRP level (Figure 6A), eosinophil count (Figure 6B), lymphocyte count (Figure 6C), or disease duration (Figure 6D). There was also no difference in IL-13 between UC with (*n* = 9) and without (*n* = 14) allergic diseases, including 5-ASA intolerance (median (IQR) 0.36 (0–1.25) pg/mL vs. 0.363 (0–2.18) pg/mL, *p* = 0.949) (Figure 6H). Furthermore, no significant difference in IL-13 with or without corticosteroids at baseline (*p* = 0.310, Figure 6I) was observed. In contrast, a trend toward a positive correlation between IL-13 and IL-4 was observed, although this was not significant (r = 0.36, *p* = 0.091) (Figure 6E). Conversely, a negative correlation was observed between IL-13 and IL-17 levels (r = −0.32, *p* = 0.135) (Figure 6G), although the difference was not significant. IL-13 and IL-15 levels were not correlated (r = −0.043, *p* = 0.847) (Figure 6F).

## 4. Discussion

In this study, IL-13 was suggested to be a potential biomarker for predicting GLM efficacy. IL-13 is produced mainly by Th2 cells, shares many physiological functions with IL-4, and is a known factor in allergy pathogenesis [13]. It has been reported that IL-13 expression is upregulated in intestinal mucosal tissues of UC, suppressing intercellular adhesion and inducing tissue damage [14]. Bram et al., reported that high IL-13Rα2 gene expression in intestinal mucosal tissues in CD is associated with the low efficacy of anti-TNFα antibodies [12]. Since IL-13Rα2 expression is enhanced by IL-13 stimulation [15,16,17,18,19], it is possible that in UC with high serum IL-13, IL 13Rα2 expression may be enhanced in intestinal mucosal tissues. In this study, GLM was less effective in bio-naïve UC patients with high serum IL-13 levels, suggesting that inflammation induced by IL-13 cannot be suppressed by anti-TNFα antibodies. In contrast, in this study, serum soluble IL-13 Rα2 was not associated with GLM efficacy, indicating that IL-13, but not soluble IL-13 Rα2, is a useful biomarker in serum.

Interestingly, a trend toward a positive correlation between IL-13 and IL-4 was observed, although not significantly different. In contrast, IL-17, which is mainly produced by Th17 cells [20], showed a negative correlation with IL-13. Both IL-4 and IL-13 are cytokines that are mainly produced by Th2 cells, suggesting a predominance of Th2 cells in UC with high IL-13 levels. Since TNF-α is thought to be mainly associated with Th1 and Th17 cells in IBD [20,21,22,23], it seems reasonable that anti-TNF-α antibody preparations are less effective in UC with Th2 cell activation.

The reason for the high IL-13 levels in UC was not clear, as IL-13 did not correlate with CRP, and IL-13 did not decrease in patients whose inflammation improved after GLM treatment. Therefore, no relationship was found between IL-13 levels and UC disease activity. Furthermore, IL-13 levels did not correlate with a history of allergic diseases. UC is known to be a heterogeneous population with various cytokine patterns [2], and some patients with UC may have high levels of IL-13; however, the cause is unknown.

In this study, serum IL-15 levels were significantly higher in the GLM response group than in the non-response group. IL-15 is produced by macrophages, B cells, and intestinal epithelial cells; is involved in several inflammatory mechanisms through both adaptive and innate immunity; and is reported to be highly expressed in the colonic mucosa in IBD [24]. In contrast, it induces the proliferation of regulatory T cells by enhancing p-STAT5 and positively regulating Foxp3 [25]. Thus, IL-15 has been shown to have both activating and regulating functions in inflammation. The results suggest that the high efficacy of GLM in patients with high IL-15 levels may be due to the anti-inflammatory effects of IL-15 mediated by regulatory T cells.

OSM has attracted much attention since it was reported that the OSM gene is highly expressed in intestinal mucosal tissues of patients with IBD resistant to anti-TNFα antibodies [11]. OSM is highly expressed in the intestinal mucosa of patients with active IBD, and as with intestinal mucosal tissue, OSM expression in serum would have been of high clinical utility if associated with the efficacy of anti-TNFα antibodies, but in this study, there was no association between serum OSM level and GLM efficacy.

This study had some limitations. First, the sample size is relatively small, which might increase statistical variability. However, the present study focused on bio-naïve UC, which reduces diversity in the clinical background. Future studies should be conducted based on a multicenter prospective design with a larger number of cases. Second, we did not analyze the expression of molecules in intestinal mucosal tissue, so we could not examine whether each molecule in serum is expressed in parallel with intestinal inflammation. Third, it is not clear whether IL-13 is a predictor of GLM efficacy in patients with UC who failed biologic therapy because this study was limited to biologic-naïve UC patients. However, the pathophysiology of UC and the efficacy of GLM are more clearly reflected in bio-naïve patients; therefore, it is important to examine the efficacy of GLM in bio-naïve patients.

## 5. Conclusions

The measurement of serum IL-13 in bio-naïve patients with UC may allow for the prediction of GLM efficacy, and thus may help to tailor personalized treatment strategies.

## Figures and Tables

**Figure 1 jcm-11-04952-f001:**
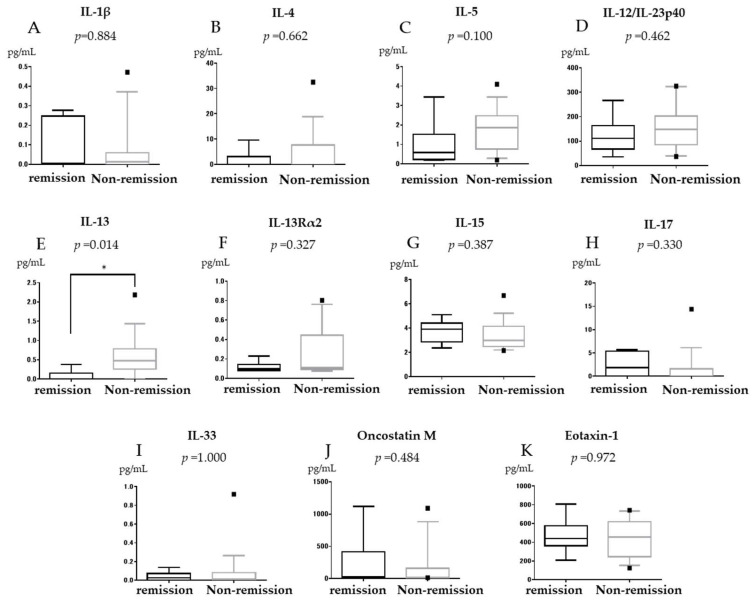
Serum Interleukin (IL)-1β (**A**), IL-4 (**B**), IL-5 (**C**), IL-12/23p40 (**D**), IL-13 (**E**), IL-13Rα2 (**F**), IL-15 (**G**), IL-17 (**H**), IL-33 (**I**), OSM (**J**), and eotaxin-1 (**K**) expression in relation to clinical remission in bio-naïve ulcerative colitis patients treated with Golimumab (GLM). * *p* < 0.05. The horizontal line inside the box represents the median. The length of the box represents the interquartile range between the first and the third quartiles. The error bars above and below the box represent 90th and 10th percentile values, respectively.

**Figure 2 jcm-11-04952-f002:**
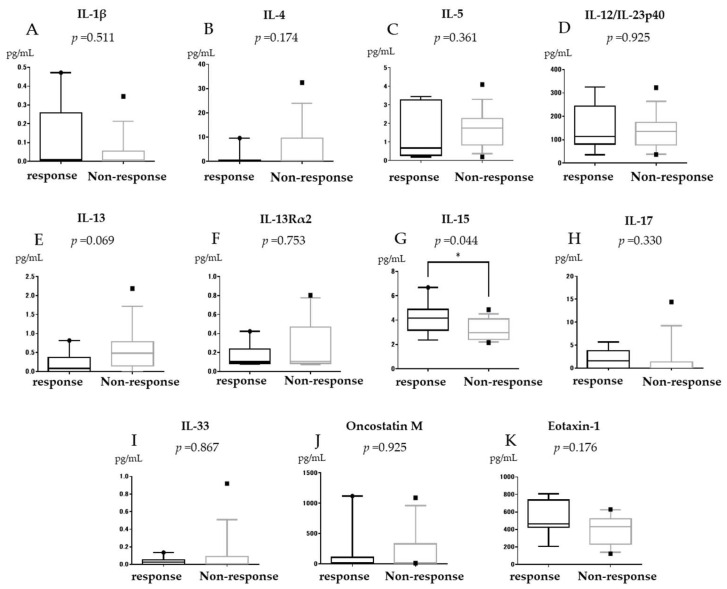
Serum Interleukin (IL)-1β (**A**), IL-4 (**B**), IL-5 (**C**), IL-12/23p40 (**D**), IL-13 (**E**), IL-13Rα2 (**F**), IL-15 (**G**), IL-17 (**H**), IL-33 (**I**), OSM (**J**), and eotaxin-1 (**K**) expression in relation to clinical response in bio-naïve ulcerative colitis patients treated with Golimumab (GLM). * *p* < 0.05. The horizontal line inside the box represents the median. The length of the box represents the interquartile range between the first and the third quartiles. The error bars above and below the box represent 90th and 10th percentile values, respectively.

**Figure 3 jcm-11-04952-f003:**
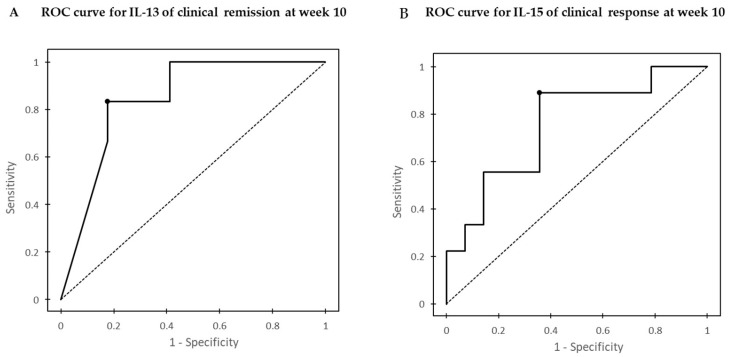
(**A**) Receiver operating characteristic (ROC) curve analysis of optimal serum IL-13 before the first GLM dose associated with clinical remission at week 10. (**B**) ROC curve analysis of optimal serum IL-15 before the first GLM dose associated with the clinical response at week 10.

**Figure 4 jcm-11-04952-f004:**
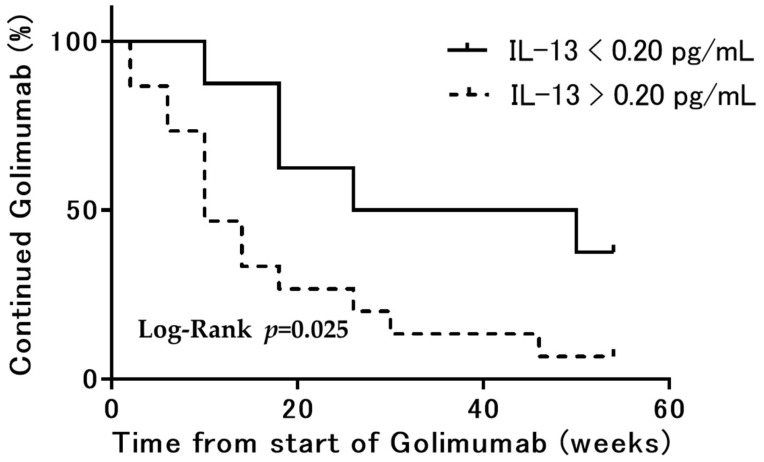
Kaplan–Meier curve analysis showing treatment termination over time for patients with serum IL-13 levels greater than 0.20 pg/mL and for patients whose serum IL-13 did not exceed the threshold.

**Figure 5 jcm-11-04952-f005:**
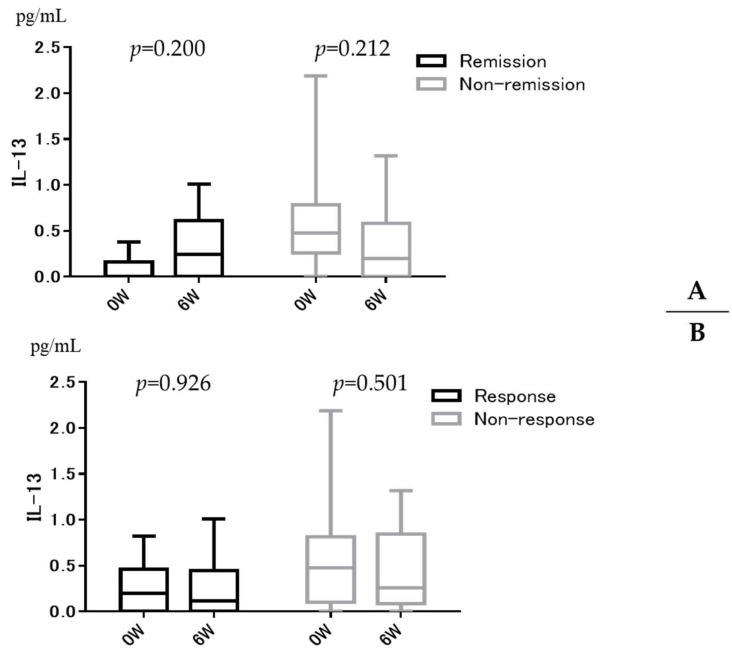
Relationship between Golimumab (GLM) remission group (**A**) or response group (**B**) and IL-13 changes over time.

**Figure 6 jcm-11-04952-f006:**
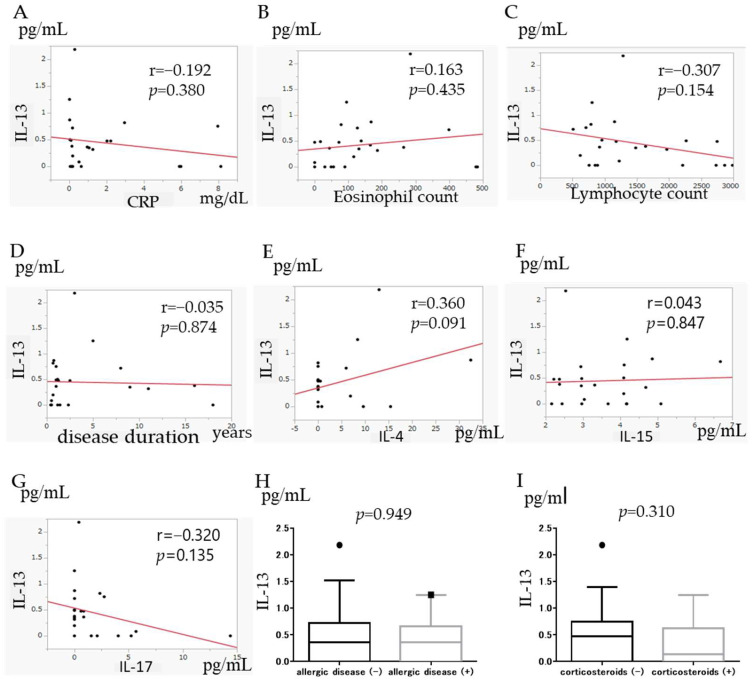
Correlation between Inteleukin (IL)-13 and C-reactive protein (CRP) (**A**), eosinophil count (**B**), lymphocyte count (**C**), disease duration (**D**), IL-4 (**E**), IL-15 (**F**), and IL-17 (**G**), and a history of allergy, including 5-aminosalicylic acid (ASA) intolerance (**H**), concomitant corticosteroids (**I**).

**Table 1 jcm-11-04952-t001:** Baseline Demographics and Clinical Characteristics.

Number of Patients, *n*	23
Male/Female, *n*	13/10
Age, year, median (IQR)	47.0 (20–81)
Weight, kg, median (IQR)	58.0 (41.9–88.8)
Duration of disease, year, median (IQR)	1.17 (0.41–18)
UC location; Left side/Extensive, *n*	6/17
Medications for UC taken at baseline	
Aminosalicylates, *n* (%)	19 (82.6)
Azathioprine, *n* (%)	6 (26.1)
Corticosteroids, *n* (%)	8 (34.7)
Corticosteroids dependent	15 (65.2)
Corticosteroids refractory	1 (4.3)
History of allegies, *n* (%)	9 (39.1)
Partial Mayo score, median (IQR)	7 (3–9)
WBC, /μL, median (IQR)	8520 (3730–15,320)
Ly, %, median (IQR)	20.8 (3.7–39.5)
Eo, %, median (IQR)	1.4 (0–4.8)
Hb, g/dL, median (IQR)	12.3 (8.8–16.2)
Platelet, 10^4^/μL, median (IQR)	33.1 (8.8–44.7)
Albumin, g/dL, median (IQR)	3.9 (2.5–4.7)
CRP, mg/L, median (IQR)	0.52 (0.01–8.17)

WBC = white blood cell; CRP = C-reactive protein; IQR = interquartile range.

**Table 2 jcm-11-04952-t002:** Baseline demographic variables.

Number of Patients, *n* (%)	Remission 6 (26.1%)	Non-Remission 17 (65.3%)	*p*-Value
Male/Female, *n*	2/4	11/6	0.341
Age, year, median (IQR)	31 (20–53)	59 (24–81)	0.023
Duration of disease, year, median (IQR)	0.835 (0.416–16)	1.25 (0.5–18)	0.247
Clinical course; Relapse-remitting/chronic continuous, *n*	6/0	13/4	0.539
Medications for UC taken at baseline			
Aminosalicylates, *n* (%)	4 (66.7)	15 (88.2)	0.27
Azathioprine, *n* (%)	2 (33.3)	4 (23.5)	0.632
Corticosteroids, *n* (%)	4 (66.7)	4 (23.5)	0.131
Corticosteroids dependent	6 (100)	9 (52.9)	0.058
Corticosteroids refractory	0 (0)	1 (5.9)	1
History of alergics, *n* (%)	2 (33.3)	7 (41.1)	1
Lichtiger index, median (IQR)	11 (9–14)	10.5 (5–16)	0.367
pMayo score, median (IQR)	6.5 (5–8)	7 (3–9)	0.639
WBC, /μL, median (IQR)	8510 (4000–15,320)	8520 (3730–13,760)	1
Ly, %, median (IQR)	15.6 (8–39.5)	20.8 (3.7–34.3)	0.916
Eo, %, median (IQR)	0.2 (0–3.8)	2.4 (0–4.8)	0.073
Hb, g/dL, median (IQR)	12.7 (11.0–14.8)	12.3 (8.8–16.2)	0.506
Albumin, g/dL, median (IQR)	3.95 (2.7–4.6)	3.75 (2.5–4.7)	0.853
CRP, mg/L, median (IQR)	0.33 (0.05–8.12)	0.64 (0.01–7.96)	0.972
Platlet, 10^9^/L, median (IQR)	38.3 (21.6–44.7)	30.0 (8.8–43.4)	0.074

## Data Availability

The data are not publicly available because there is no appropriate site for uploading at present. The data underlying this article will be shared upon reasonable request to the corresponding author.

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
