# Peer review of "Serum IL-13 Predicts Response to Golimumab in Bio-Naïve Ulcerative Colitis"

_jcm, 2022, doi:10.3390/jcm11174952_

Round 1

Reviewer 1 Report

This is an interesting study of predictive parameters of golimumab responsiveness/remission in bio-naive patients with ulcerative colitis.

-one drawback is the small sample

-p.2 abbreviation ECL should be explained

-there are some misspellings on page 4 as dependent..

-there is a significant mistake in the results section p.5 line 138: relation should be reverse as for remission

-all the remission patients were steroid dependent which leads to some further questions:

-what was the steroid dose at baseline and in week 10, week 52 in both groups?

-was there any interference of steroids on IL13?

-what is known  about IL13 and steroids in the literature?

Over all the negative effect of high IL13 was consistent on remission, response and GLM discontinuation in these patients and therefore these results are credible although they should be confirmed in a bigger cohort.

Reviewer 2 Report

Define: sensitivity, pMAYO score, how is this different than MAYO score?

The paper is well written an brings up a very  important issue.  While this is an important research that needs to be published, my major issue/concern is that N value might be too low to drive a conclusion n=2 Males/ 4 females given the variability seen in the figures.

add comments: My decision was based on the fact that there was a much lower n number to justify the conclusion. While this seems like a minor point, and it could be, if the authors can include more data points to rectify the issue, this to me at least for now seemed like the only flaw and a "major" issue with the paper, which precludes us from driving the conclusion. As it stands the remission group only has 2 males and 4 female patients, and the variability is too high in my opinion to run the stats (see fig 1 and 2). If the authors can come back and convince me that the points I raised are invalid, I would be more than happy to take back my recommendation. Overall, I like the paper, it is well written and the responders vs non responders research is of considerable area of interest.

Round 2

Reviewer 2 Report

Thank you for addressing the issue that I raised head on. Great to see that   the limitations of the study and future recommendations have been added to the manuscript.